# Why do people take part in atrial fibrillation screening? Qualitative interview study in English primary care

Sarah Hoare ®,[1] Alison Powell ®,[1] Rakesh Narendra Modi ®,[2] Natalie Armstrong ®,[3] Simon J Griffin ®,[2,4] Jonathan Mant ®,[2] Jenni Burt ®,[1] The SAFER Authorship Group

[1]The Healthcare Improvement Studies Institute, Department of Public Health and Primary Care, University of Cambridge, Cambridge, UK
[2]Primary Care Unit, Department of Public Health and Primary Care, University of Cambridge, Cambridge, UK
[3]Department of Health Sciences, University of Leicester, Leicester, UK
[4]MRC Epidemiology Unit, Institute of Metabolic Science, School of Clinical Medicine, University of Cambridge, Cambridge, UK

**Correspondence to**
Dr Sarah Hoare;
sarah.hoare@thisinstitute.cam.ac.uk

## ABSTRACT

**Objectives** There is insufficient evidence to support national screening programmes for atrial fibrillation (AF). Nevertheless, some practitioners, policy-makers and special interest groups have encouraged introduction of opportunistic screening in primary care in order to reduce the incidence of stroke through earlier detection and treatment of AF. The attitudes of the public towards AF screening are unknown. We aimed to explore why AF screening participants took part in the screening.

**Design** Semistructured longitudinal interview study of participant engagement in the SAFER study (Screening for Atrial Fibrillation with ECG to Reduce stroke). We undertook initial interviews face to face, with up to two follow-up telephone interviews during the screening process. We thematically analysed and synthesised these data to understand shared views of screening participation.

**Setting** 5 primary care practices in the East of England, UK.

**Participants** 23 people taking part in the SAFER study first feasibility phase.

**Results** Participants were supportive of screening for AF, explaining their participation in screening as a 'good thing to do'. Participants suggested screening could facilitate earlier diagnosis, more effective treatment, and a better future outcome, despite most being unfamiliar with AF. Participating in AF screening helped attenuate participants' concerns about stroke and demonstrated their commitment to self-care and being a 'good patient'. Participants felt that the screening test was non-invasive, and they were unlikely to have AF; they therefore considered engaging in AF screening was low risk, with few perceived harms.

**Conclusions** Participants assessed the SAFER AF screening programme to be a legitimate, relevant and safe screening opportunity, and complied obediently with what they perceived to be a recommendation to take part. Their unreserved acceptance of screening benefit and lack of awareness of potential harms suggests that uptake would be high but reinforces the importance of ensuring participants receive balanced information about AF screening initiatives.

**Trial registration number** ISRCTN16939438; Pre-results.

---

**Strengths and limitations of this study**

► Our research adds to the limited evidence base about atrial fibrillation (AF) screening participation: opportunistic AF screening is encouraged in primary care to reduce the incidence of stroke through earlier detection and treatment of AF, despite insufficient evidence currently available to support systematic AF screening.

► We report the views of people taking part in AF screening as part of a research study, limiting the relevance of our findings for understanding public engagement in either opportunistic AF screening conducted as part of routine primary care or potential future systematic AF screening programmes.

► This was a study of people who participated in AF screening and it does not address the views of those who were invited and opted to not take part.

► The lack of ethnic diversity in our sample reduces the utility of the results, particularly because attitudes towards screening and AF are known to differ by ethnic background.

► Our research contributes to social science literature about the public's 'moral obligation' to participate in screening even when both the programme and the condition are largely unfamiliar to participants.

## INTRODUCTION

Opinion on whether there should be a screening programme for atrial fibrillation (AF) is divided.[1–3] Screening advocates include clinical societies,[4–7] patient associations,[8] clinicians,[9 10] and pharmaceutical and technology companies.[11 12] Justifications for AF screening are well rehearsed. Having AF (a common, often asymptomatic, heart arrhythmia) increases the risk of stroke five-fold[13] with AF-related stroke typically more severe than non-AF-related stroke.[14] AF anticoagulant treatment is effective at reducing stroke risk,[15] and screening devices for use in the community are inexpensive.[2] Conversely, caution around AF screening is attributed to insufficient evidence on effectiveness,

cost-effectiveness and potential harms of national systematic AF screening programmes, which are not currently recommended.[16 17] Questions remain, for example, about the difference in stroke risk and anticoagulation benefit for people with AF detected by screening rather than in routine care.[16 18]

In the meantime, opportunistic AF screening is encouraged in primary care, leading to accusations of 'back-door' screening.[12] Opportunistic screening occurs through many routes. In the UK, clinicians are recommended to assess patients' pulse rhythm as part of National Health Service (NHS) health checks,[19] driven by an explicit aim of NHS programmes and policy to increase identification and diagnosis of previously undetected AF.[20 21] NHS England has distributed digital ECG devices to increase AF detection,[22] and some clinical commissioning groups have encouraged opportunistic AF case-finding in primary care.[23 24]

The public's views have largely been absent from this debate. Patient associations which champion AF screening provide important information from the perspective of a motivated minority, but tell us little about the opinions of the wider public.[12] If typically positive attitudes to screening programmes apply here,[25 26] it seems likely that the public will be enthusiastic about AF screening, as seen in a small-scale study of AF screening trial participants[27] and informal reports from a previous AF screening trial.[28] However, little is known about the reasons why the public are motivated to participate in AF screening.[27 28] We aimed to explore why participants in Screening for Atrial Fibrillation using ECG to Reduce stroke (SAFER), an AF screening study, opted to take part.

## METHODS
### Design and participants
Our data are from a longitudinal interview study with participants in the AF screening study SAFER (https://www.safer.phpc.cam.ac.uk/). We interviewed participants during their study involvement to explore their experience of AF screening, including their views on why they had taken part.

The SAFER programme is ongoing, and interviews were conducted as part of a feasibility study. Participants were drawn from five participating general practitioner (GP) practices in the East of England; eligible patients were aged 65 years and over, not taking anticoagulant medication, and were neither on a palliative care register nor living in a residential (care) home. Participants were first invited to contribute to SAFER research and subsequently to take part in AF screening. The screening involved using a Zenicor (www.zenicor.com) hand-held single-lead ECG device four times a day for between 1 and 4 weeks.

We selected interview participants from the first wave of feasibility phase participants. Potential interviewees were purposively sampled by age, gender and GP practice to ensure a varied sample. All participants were invited by

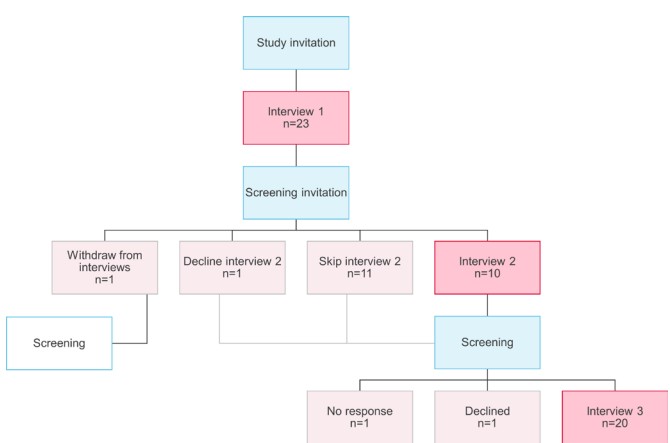

**Figure 1** Flow chart of interviews in the screening process.

letter and agreed to take part in SAFER and, on invitation to screening, elected to take part.

### Data collection
We interviewed participants up to three times throughout their screening. Interview one occurred shortly after participants agreed to take part in the SAFER research study; interview two after a participant had received their invitation to take part in SAFER screening; and interview three after they had completed the screening. Interview one was held either at the participant's home or at their GP practice, and interviews two and three were conducted by telephone. Not all participants took part in all interviews: participants could elect not to take part in each interview phase, and some participants were not invited to the second interview to ensure that they had the opportunity to participate in screening (see figure 1). Interviews were typically held with only the interviewer (SH) and interview participant, but when participants' partners were present they were included in the discussion and consented to take part. Fieldnotes were completed following each interview.

Interviews were semistructured and used a flexible topic guide exploring experiences of, and attitudes towards, screening in general and AF screening in particular (see online supplemental material). These topics were explored by participants throughout all three interviews. The topic guide was designed to reflect our research question and was refined and iteratively adapted as interviews progressed to accommodate areas of interest raised by participants. Consent was taken prior to each interview. Interviews were conducted in 2019 and continued until we had reached sufficient information power[29] to have a meaningful dataset. Interviews lasted on average 30 min each (range 10–90 min) and were audio recorded. We completed 53 interviews with 23 participants (interview 1 n=23, interview 2 n=10, interview 3 n=20). Table 1 lists participants' characteristics.

### Analysis
Interviews were transcribed verbatim and analysed using a thematic approach to explore the reasons why people

| Table 1 | Sociodemographics of participants (n=23) |
| --- | --- |
| **Participant characteristic** | *n* |
| Age group | |
| 65–69 years | 9 |
| 70–74 years | 7 |
| 75–79 years | 5 |
| 80+years | 2 |
| Sex | |
| Female | 10 |
| Male | 13 |
| Practice (deprivation score 1–10; 10=least deprived)* | |
| A†(10) | 8 |
| B (8) | 5 |
| C (8) | 4 |
| D (6) | 4 |
| E (5) | 2 |

*The deprivation score is taken from National General Practice Profiles,[73] using the English Indices of Deprivation to calculate the Index of Multiple Deprivation, which provides "an overall measure of deprivation experienced by people living in an area".[73 74]
†Practice name pseudonym.

took part in AF screening. Interviews were collated, and the initial analysis phase focused on 16 transcripts selected for their relevance to addressing the question and drawn from all three interview timepoints (interview 1, 7 transcripts; interview 2, 3 transcripts; interview 3, 6 transcripts, with some participants represented more than once). This initial analysis was followed by analysis of the whole interview dataset (n=53). SH conducted coding, on paper and then supported by the software NVivo V.12, with codes generated inductively from topics raised by participants and interview fieldnotes, and deductively from the interview schedule. We developed key themes through consensus meetings between SH, AP and JB, and SH then explored these themes within the wider dataset to establish the veracity of key themes and identify deviant cases, with the themes subsequently refined. These themes were synthesised to understand shared views of screening participation, aided by reference to social science and health screening literature about participation in screening.[30–38]

## Patient and public involvement (PPI)

The SAFER programme is guided by four PPI representatives[39] who have been involved since the programme was conceived and funded. The representatives are embedded within the team (eg, one is a coapplicant, another sits on the programme steering committee) and advise on all aspects of the design, management and delivery of the programme, including interpreting and disseminating the findings. We have also established a supplementary qualitative work-stream PPI group with whom we regularly consult about research design questions.

## RESULTS

Quotations in the text are followed by the participant's interview ID number (1-53), practice code (A-E) and interview phase (1-3).

### Summary

Engagement with AF screening within the SAFER study was driven by several interconnected considerations, many relating to wider attitudes that screening is a 'good thing' to participate in, and to the perceived importance of preventive health behaviours. Participants knew little about AF, but were motivated to be screened by the perceived potential for reducing their risk of catastrophic disability associated with stroke, and a seemingly low-risk screening test with few if any apparent harms and significant perceived benefits.

### The importance of screening in general

Participants were familiar with national screening programmes, often referring to their own or a relative's prior experience of taking part in screening. Screening was described as a 'no-brainer' by several participants, and it was clear within our sample that screening was seen as a routine healthcare activity in which participation was the only reasonable option. This option was driven by a desire to follow clinical recommendations, gain insight into health status and, if required, access early treatment, and was further underpinned by a 'fear of missing out' by not participating.

#### Clinically recommended screening

Screening invitations were typically seen as legitimate requests from a clinical authority which was presumed to recommend the screening programme. Participants explained that this meant that screening was something they ought to at least consider engaging with, and for a minority as a mandate that they must comply with.

> And I assume when something like that [a screening invitation] comes from the post, I immediately comply, I don't question. I don't question because it's come from a…you know, it's a bit different to somebody trying to ring me up and telling me that they can spend my pension more successfully than I can. It comes from an authoritative source, and I don't think there's any question of me not believing what the prospects are. (05B_1)

#### Reassurance of screening

Participants expected screening would provide them with reassurance about their health. This could be definitive, with a negative test result providing 'peace of mind' that the screened for condition was 'something else you can tick off the list of things to worry about' (06E_1). Reassurance could also be more anticipatory, whereby participation in screening was protective of regret if in the future one did indeed get diagnosed with the condition but had chosen not to be screened. Simply attending screening also seemed to confer reassurance about healthiness,

whereby participants additionally acknowledged responsibility for maintaining their health and preventing ill-health:

> But, I think, it's a brilliant idea and the thing is though, that things can happen in between, so you've still got to be responsible for your own health. (25A_1)

### Early diagnosis, early treatment, better outcomes

Participants were clear that early identification of a screened condition would enable preventive action or prompt treatment to avert ill-health or reduce disease severity. Participants placed different emphases on the importance of these potentialities (from prevention to treatment) but were united by a presumption that earliness was advantageous. By comparison, 'late' identification or treatment was associated with perceived poorer outcomes for individuals and the healthcare system. Participants often drew on this binary conceptualisation of early versus late identification to explain why it was sensible to take part in screening:

> I'd rather find something out, if there is a screening thing that can indicate early symptoms that can be caught early and something can be done about it, I think I'd much rather have that option than sort of find out later, you know, further down the road that, well it's too late now. (16D_1)

### *Identifying silent conditions: robust test*

Underlying perceived benefits of early diagnosis was participants' awareness that screening could identify a condition before symptoms could be noticed or manifested, or for which symptoms were absent (a 'silent' condition):

> I get this thing for the bowel screening, which isn't as much fun. And despite that, I've gone in for it, because as a layman, I simply have no idea what's going on inside my body. The fact I feel fit, doesn't mean I am fit, you know, from those sort of complaints. [I: So…] So the answer is, I've said yes to that [bowel cancer screening] and I would have said yes to this [AF screening]. (11A_2)

The ability of screening to identify silent conditions led to a view among some participants that screening tests could provide a particularly in-depth or robust review of health. For a few, this extended the scope beyond the screened condition; here, the SAFER AF screening came to be viewed as a 'free medical' (15A_3). Screening thus offered ready access to healthcare and perceived 'health checks' without the need to present tangible health symptoms:

> R: "I think, I don't want to bother the doctor he's so busy, so I won't bother to go until I really think, I'm going to have to go. […] I think, a lot of people don't go, so screening, perhaps, would ensure that they are

seen for a particular thing, once every three years or however they're done."

> I: "And, do you think going to a screening appointment is different to just going for something else?"

> R: "Well, I think, it has to be 'cause I don't think anybody, unless they've got any symptoms, has any reason to think they've got anything." (25A_1)

### *Not taking part*

Associated with reducing the risk of 'late' diagnosis, participating in screening also protected participants from moral judgement about *not* engaging in screening. Non-screeners were variously portrayed by participants as unduly embarrassed, uninformed, indolent, irresponsible, wilfully ignorant, or gratuitously anxious, and often a combination of these. A significant part of participants' moralising about non-screeners was concern that they were missing out on the benefit of screening:

> I mean maybe there are some people that don't want to know. They live in a dream world and…but I just think if you've got the chance to do something about it and it gives you a better quality, even if it's only a short one, that's great, you should grab it." (15A_3)

### *AF screening*
### Views about AF

AF was a new condition to most participants, except those with first-hand experience as a result of a family member's diagnosis or prior clinical knowledge. Some participants struggled to pronounce the term and many explained that they had conducted internet research to better understand the condition, supplementing the study literature provided. Despite this, after interviewer explanation, participants engaged with the concept of an irregular heartbeat and described what they thought it might be like to have the condition. At best, participants understood AF as a benign condition, but most expressed concern about the association with stroke. Participants anticipated that, if they had AF, they would worry about having a major and sudden heart-related incident, and expected having to modify their lifestyle (for some to a very significant and limiting degree) and to receive corrective treatment to address it:

> I: […] "I wonder what you thought it might be like to have an irregular heartbeat?"

> R: "A little bit disconcerting. I would say you would want it regulated, maybe if it's to have a pacemaker fitted or something. You wouldn't want to live with something that you know…you'd probably be able to feel it beating faster. You wouldn't want to think to yourself, am I having a stroke shortly? So I would say it's more peace of mind to know that you can regulate it if you have a… I mean, if you know that your heart is beating faster and it could be a start of a stroke, and you can take a pill to stop it and you didn't know

about it, you'd be a bit miffed to say if you didn't have one." (06E_1)

### Views about AF screening

Most participants did not expect to receive a diagnosis of AF as a result of taking up the offer of screening. They explained this in relation to their own perception of their health, whether because they were healthy overall, or because they identified an absence of AF symptoms or prior heart-related issues. Despite this, participants explained that participating in screening was useful because there was a *chance* that they may unknowingly have AF. As a silent condition with perceived serious consequences and a presumption for curative treatment, participants recognised the utility of checking to see if they had it:

> I: "Do you think the atrial fibrillation screening is relevant to you?"
>
> R: "I don't suffer from any of those symptoms. That's as far as I can…or I'm not aware of suffering from those symptoms. That's as far as I can go. Like a lot of things that are more evident with age, I think it ill behoves me to say, oh, I'm, you know, I don't need to be tested because I feel fine." (11A_2)

### Stroke fear

Participants were clear that stroke was not a condition they wished to experience. The consequences of stroke were recognised to be potentially debilitating and life-changing. While those who knew people who had had a stroke often recognised a diversity of recovery experiences that included more positive outcomes, there remained a concern about the high level of disability stroke could cause.

Participants typically recognised both that a stroke could happen unexpectedly 'to anybody at any time, any day, any age really' (16D_1) and that there were genetic and modifiable lifestyle factors which could increase the risk of having a stroke. Together with concern about the consequences of having a stroke, this dual conception of stroke as preventable and unexpected made the association between AF and stroke significant, and thus AF screening particularly worthwhile:

> I: "What do you think about the link between atrial fibrillation and stroke?"
>
> R: "I would try to do everything I can not to have a stroke. So if there was something I could do that would stop me turning into a vegetable, I will do it. So as far as I'm concerned, it's a no-brainer. Once you've had the stroke, whatever life you have got left, it's not going to be much fun, so try and avoid it." (06E_1)

### Low-risk screening

The AF screening test involved participants placing their thumbs or two fingers on a portable ECG device. Participants unanimously recognised this to be non-invasive and typically understood the screening to be safe and to not cause harm, often aligning it with the ease of measuring blood pressure.

Participants acknowledged that anxiety around the screening test may arise for others. However, they frequently related this to the screened individuals' psychological state rather than something inherent to the testing process. Participants rarely recognised other iatrogenic screening harms. If they did, such harms were presented as legitimate concerns and often as intrinsic to testing. However, participants presumed the screening was safe and, when discussed, positioned consideration of these issues as the responsibility of the screening provider.

Compared with the perceived fuzzy and inconsequential harms of screening, participants saw the advantages of AF screening as tangible and significant. Screening could identify AF 'early', using a test that was perceived to cause no harm, and which could ameliorate the risk of stroke:

> R1: "Any harm? No I don't think that any harm can arise from that all, personally, screening, no."
>
> R2: "I mean to say, it looks as if just by putting your thumbs on the actual screening test on the box, you're not going to get anything invasive from that. And, I say, if it shows up something that can be dealt with sooner rather than later, that can only be a good thing in our view really." (16D_2)

## DISCUSSION
### Summary

We explored the reasons why people took part in AF screening through interviews with SAFER study participants. Their accounts presented screening as a routine obligation and something one 'ought' to do as a responsible patient to ensure good health and prevent illness. Limited awareness of AF did not detract from their view of the utility of AF screening—rather, preventing stroke was a strong rationale for participating. Participants assessed AF screening to be a legitimate, relevant and safe screening opportunity, and complied obediently with what they perceived to be a recommendation to take part.

### Strengths and limitations

Our research adds to the limited evidence about AF screening participation. Our findings also contribute to social science literature about patients' 'moral obligation'[30] to participate in screening,[30–33] even when both the programme and the condition are largely unfamiliar to participants.

It is likely that our results reflect the delivery of SAFER as a research study. While the SAFER study has been designed to mimic a national screening programme, participants were aware that it was research. This may limit the relevance of our findings for understanding public engagement in either opportunistic AF screening conducted as part of routine primary care, or potential future systematic AF screening programmes. For example, participants reported that a key reason for taking part was

to 'help out', a prevalent and well-recognised motivation for participating in research.[40]

This was a study of people who participated in AF screening and does not address the views of those who were invited and opted to not take part. Nevertheless, focusing on people who did engage has allowed us to understand the information needs and potential misconceptions of this population.

Participants' familiarity with, and participation in, national screening programmes suggests their enthusiasm for AF screening will not be relevant to all screening invitees. For example, recent studies in the UK found that prior attendance in screening is positively correlated with future participation.[41 42] Participants in our study match national trends by being less deprived by practice deprivation status.[43] The lack of ethnic diversity in our sample will also necessarily reduce the utility of the results, particularly because screening uptake and attitudes towards AF differ by ethnic background.[43 44]

### Comparison with existing literature

Participants' concern about having a stroke mediated their unfamiliarity with AF and encouraged them to take part. The novelty of AF is unsurprising as studies find limited awareness of the condition, whether the participants' relationship to AF is: just at risk;[45] being screened for it;[27] prior to diagnosis;[46] or having it.[44 47] Similarly, those with AF report being worried or fearful about their condition,[46 47] consider it a serious heart disease[48] and overestimate the risk of stroke.[49] AF was explained to participants in the context of stroke risk, and this may explain participants' worries about AF-induced stroke: the majority of AF patients in a 2002 study[44] did not consider their condition to be severe, but almost half were not aware that having AF predisposed them to stroke.

The enthusiasm participants had for AF screening, and the weighting they attributed to the benefits versus the harms of screening, accord with other studies on the public's experience of national screening programmes.[50 51] There is widespread public support for screening even in survey scenarios in which there was no treatment for the screened condition,[25 26] or if screening is not clinically recommended because participants are outside screening age thresholds.[52 53] The benefits SAFER participants attributed to AF screening concur with reported experience. Screening offers the hope of reassurance[30 34 35] and confirmation of healthiness,[34] or in the worst case, the benefit in knowing one's ill-health status[54] and associated advantageous early diagnosis or treatment.[55] Participation also offers an 'insurance policy',[56] whereby the often presumed 'health check'[35] means those taking part can be assured that 'they had done everything they could'[34] if they later develop the screened condition. Like SAFER participants, this enthusiasm is contextualised by reported limited concern or awareness of screening harms.[55] A recent US study found that many respondents could not name any harms of screening, and where they could, focused, as SAFER participants and other AF screening

participants did,[27] on the direct harms of the screening test itself.[57] Even when harms are recognised, these are typically of a lower priority to participants compared with the benefits of screening.[58]

Informing participants' consideration of the harms and benefits of screening was an assumption that taking part in screening was something they ought to do, corresponding with evidence from screening decision-making studies.[53 59–62] Sociological work has shown how screening participation is associated with maintaining, and being responsible for, one's own health,[30–38] and part of patients' efforts to be a 'good patient' and to use healthcare resources appropriately.[63]

### Implications for practice and research

The public can be expected to be unfamiliar with AF, to anticipate that screening is recommended, to perceive any AF test as a comprehensive heart function review and to be positive about screening. Clinicians should be prepared to work with members of the public interested in screening to help them understand what they are being offered and what the risks and benefits of taking part are. SAFER participants suggested that early identification leads to better results. Counselling individuals on the benefits and risks of AF screening is particularly pertinent given participant expectations for curative treatment and because clinical evidence on the benefit of managing screen-detected AF is still uncertain.[18] As with all screening, benefits must be weighed against harms.[64]

Receiving a positive result may be unexpected for those engaging in screening for health confirmation, and could engender feelings of health vulnerability.[65] These participants may need help to moderate concerns about the severity and lifestyle impact of the condition. Clinician engagement is significant because their reactions and approaches to AF have been found to impact patients' perceptions of their condition,[66] and patients' understanding of AF may be aided by often called-for patient education interventions.[44 49 67 68] Negative test results may confer 'healthiness'[69] and induce false reassurance about their stroke risk. Though the evidence base is small,[70] it is plausible that participants' existing 'unhealthy' behaviours are validated as acceptable and continued,[71] while any future AF-related symptoms may be 'downplayed',[72] discouraging prompt healthcare seeking. Consequently, clinicians are recommended to contextualise negative results with reference to the necessary limitations of any screening test and participants' ongoing risk of developing the condition.[69]

Our study has contributed to a small evidence-base on public experience of AF screening.[27] If AF screening is demonstrated to be effective, then complementary studies would be important to understand the attitudes of people who do not participate in screening and how participants (particularly those with positive results) weigh up the benefits and harms throughout the screening process.

**Acknowledgements** We thank the SAFER participants for their time and for sharing their experiences of taking part in screening. We also thank the GP practices involved and the local National Institute for Health Research Clinical Research Network for supporting this study. We are grateful to the SAFER study team and in particular to Millie Watson for her work in administering these interviews, and to James Brimicombe for managing the electronic SAFER data.

**Collaborators** The SAFER Authorship Group includes: Andrew Dymond, University of Cambridge, UK; Richard Hobbs, University of Oxford, UK; Rachel Johnson, University of Bristol, UK; Richard McManus, University of Oxford, UK; Kate Williams, University of Cambridge, UK.

**Contributors** JB planned the interview study and oversaw the data collection by SH. SH led the analysis with themes developed with JB and AP, and contributions from RNM. SH and JB drafted the original manuscript. SH, AP, RNM, NA, SJG, JM, JB critically reviewed, revised and approved the final version. All members of the SAFER Authorship Group (AD, RH, RJ, RNM, KW) reviewed and approved the initial submission and had the opportunity to comment, and the final version was shared with them. JB is the guarantor.

**Funding** The SAFER study is funded by the National Institute for Health Research (NIHR) Programme Grants for Applied Research (grant reference number RP-PG-0217-20007) and School for Primary Care Research (SPCR-2014-10043, project 410). The views expressed are those of the authors and not necessarily those of the NIHR or the Department of Health and Social Care. RNM is supported by the Wellcome Trust as part of the Wellcome Trust PhD Programme for Primary Care Clinicians (grant reference number 203921/Z/16/Z). NA is supported by a Health Foundation Improvement Science Fellowship (N/A award/grant number) and also by the National Institute for Health Research (NIHR) Applied Research Collaboration East Midlands (ARC EM) (N/A award/grant number). The views expressed are those of the authors and not necessarily those of the NHS, the NIHR or the Department of Health and Social Care. The University of Cambridge has received salary support in respect of SJG from the NHS in the East of England through the Clinical Academic Reserve (N/A award/grant number). JM is an NIHR Senior Investigator (N/A award/grant number). SH, AP and JB are based in The Healthcare Improvement Studies Institute (THIS Institute), University of Cambridge. THIS Institute is supported by the Health Foundation, an independent charity committed to bringing about better health and healthcare for people in the UK (N/A award/grant number).

**Competing interests** JM has received honoraria from BMS/Pfizer.

**Patient consent for publication** Not applicable.

**Ethics approval** This study involves human participants and was approved by London-Central NHS Research Ethics Committee (reference number: 18/LO/2066) Participants gave informed consent to participate in the study before taking part.

**Provenance and peer review** Not commissioned; externally peer reviewed.

**Data availability statement** Data are available upon reasonable request. Please email the PI Jenni Burt (jenni.burt@thisinstitute.cam.ac.uk) for details.

**ORCID iDs**
Sarah Hoare http://orcid.org/0000-0002-8933-217X
Alison Powell http://orcid.org/0000-0003-2524-5357
Rakesh Narendra Modi http://orcid.org/0000-0001-9651-6690
Natalie Armstrong http://orcid.org/0000-0003-4046-0119
Simon J Griffin http://orcid.org/0000-0002-2157-4797
Jonathan Mant http://orcid.org/0000-0002-9531-0268
Jenni Burt http://orcid.org/0000-0002-0037-274X

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
