## [Reviewer comments · BMJ Open]

ARTICLE DETAILS

TITLE (PROVISIONAL)	Why do people take part in atrial fibrillation screening? Qualitative interview study in English primary care
AUTHORS	Hoare, Sarah; Powell, Alison; Modi, Rakesh; Armstrong, Natalie; Griffin, Simon; Mant, Jonathan; Burt, Jenni; SAFER Authorship Group, The

VERSION 1 – REVIEW

REVIEWER	Gwynn, Josephine University of Sydney - Mallett Street Campus, charles perkins centre
REVIEW RETURNED	21-Jun-2021

GENERAL COMMENTS	METHODS: Data collection : Page 7 line 23-26: Please provide more details regarding the prompts used during interview. Only a general statement is provided. It would be appropriate to add the interview prompt schedule as an appendix, particularly as in the analysis section you refer to the themes being developed deductively from the interview schedule. It's therefore important we have more information about that schedule, to this understand the basis of the themes that are presented. Analysis: Page 7 Line 32: provide an explanation for the statement that '16 transcripts were selected for their relevance'. What do you mean by relevance please define? Page 7 Line 41: figure one. The figure does not transparently report at which point across the three interviews the final 16 were selected. Were the same number selected at each interview point? Or were there different numbers at different interview points? Arguably the patient experience is different at each interview point and this may have influenced their experience and the information they provided. Page 7 line 39: you refer to the themes being synthesised and aided in this process by reference to social science and health screen literature. Please provide references for the literature you used as the basis for managing the themes. Patient and public involvement: Page 7 line 46: here there is a term 'PPI representatives'. I can't see that defined previously if it hasn't been, please provide that here. The term may be particular to the UK. Results.
---

	Page 7 line 58: here the authors refer to completing 53 interviews. However previously in the analysis section you state that you completed 16 transcripts/interviews. This is very unclear. I note later in the analysis section you refer to the 'rest of the data set'. If you mean that the data set was not the 16 transcripts but was the wider 53 interviews, then this needs to be much clearer in the analysis. Figure 1 should also reflect this process. Page 7 line 58 and 59: please provide more demographic information Regarding the participants under a subheading labelled demographics . The inclusion of the results at this point is a little unclear, please provide another subheading to introduce the sentence / section beginning with 'engagement with AF screening'. The sentence from page 7 line 59 through to page 8 line 7 seems to be related to the following subheading on the importance of screening. It might be relevant to merge the two. Discussion. Page 11 line 58 to 60 and 12 line 3 to 4: this is an important discussion point and speaks to the relevance of this paper. The debate for AF screening is particularly strongly based around the need to screen and therefore engage hard to reach and vulnerable populations. These populations are not included in this study, It will be highly relevant to include some discussion regarding the learnings from this study as applied to the hard to reach / vulnerable populations. Can the factors that drive an already engaged (and relatively privileged group going by the demographics) inform engagement of hard-to-reach populations? I think the manuscript would be greatly enhanced by expanding on this point and truly contribute to the discussion on the broader issue of AF screening. Page 13 line 3: '...because clinical evidence on the benefit of managing screen detected AF is still uncertain'. This phrase speaks to the debate against screening in that screening can induce anxiety additional requirements for medical appointments and treatment and particularly amongst underserved populatio
--	---

REVIEWER	Salmasi, Shahrzad Collaboration for Outcomes Research and Evaluation (CORE), Faculty of Pharmaceutical Sciences, Faculty of Pharmaceutical Sciences, The University of British Columbia
REVIEW RETURNED	07-Jul-2021

GENERAL COMMENTS	Abstract: Under design n is stated as 53 but under results n is reported to be 23. In the manuscript it becomes clear that 53 is the number of interviews and 23 is the number of participants (multiple interviews per patient) but this is not clear in the abstract and is confusing. Objective is misleading: "We aimed to explore why people take part in AF screening". You aimed to explore why people who took part in AF screening did so.
--

	Under design, "Interviews explored participants' experience of and attitudes towards AF screening." Is reiteration of the objective. Suggest using this space to provide more detail on the methods of the paper. Results is misleading: "participants were supportive of screening for AF, describing it as a "good thing to do". In those who took part in screening, one of the reasons stated was that it is a good thing to do. Introduction Reads well and is appropriately cited. Methods Please make sure all abbreviations are explained the first time they appear in the manuscript even the commonly used ones such as ECG and GP. Please provide more information on the steps involved in thematic analysis. If word limit is the issue, such details can be provided as appendix. Please attach your coding tree or code book as appendix. You cannot use the word patient. Participants in your study and all those who will ever go for screening are not patients because they have not yet been diagnosed. This happens often in the manuscript e.g: "Patients can be expected to be unfamiliar with AF", "eligible patients were aged 65 years and over, not taking anticoagulant medication, and were neither on a palliative care register nor living in a residential (care) home". Participant, the public, and adults, are good replacements. Please correct throughout the manuscript. Please provide a sample of the interview guide in the appendix More information is required on the methods:
--	---

	when did interviews stop? Was the sample size prespecified or did the interviews continue until the point of saturation? How was saturation defined? How was the interview guide designed? Based on author's clinical experience? Review of the literature? Adaptation of an interview guide from another study? Was there a pilot phase to improve the interview guide? And the interview process? Was the interview guide revised iteratively as more interviews were conducted? Was the analysis done iteratively or at the end? And why. What was the reason for interviewing participants 3 times. You have not assessed changes in perception/understanding/willingness of participants throughout time, and have treated the interviews as if they are each independent of one another, so why the longitudinal design? Results Actual mean interview duration must be reported. It is not clear what the codes in the brackets represent. Assuming they refer to the participants ID, I am not sure if having them would help the reader in any way to better contextualize or understand the quote. Basic information such as (Male, 65) may be more helpful. Quotations are usually italicized. I refer to the editors on if they should be in this paper. It is not clear which of the titles represent the themes and which are the subthemes (if there is any). Numbering the themes usually helps with this. Some of the titles are in red, I am not sure why. Are these the main themes? It is very likely that they will all show up as black in the final version so this is not the best way to distinguish themes from subthemes. Theme names should be more descriptive and meaningless even without the text. Titles such as "low risk" , "screening", "AF" are not helpful. The point of reporting guidelines such as COREQ is to ensure standardization of reporting and to ensure that all the important information about a study has been reported. You cannot have NA
--	---

in your checklist, all information in COREQ must be reported in your manuscript.

I am unable to find information on field notes, information on how many data coders coded the data and information on interviewer's relationship with the participant, even though the authors indicate having reported them in the COREQ checklist. Page numbers change throughout the publication process, instead of page numbers it might be helpful to provide the title/subtitle under which the info appears. For example page 7 is supposed to have information on whether the guide was pilot tested but this information is not there.

I am not sure if dividing the results on screening in general and AF screening specifically is a good idea. It is confusing, at times feel repetitive.

The fact that the people who were interviewed are those who volunteered to join a screening program, needs to be emphasized throughout the paper and more clearly in discussion. It is not so much an issue that they were part of a prior research study but that they were part of a screening research study.

Results is written as if these are perceptions of general public, they are not. They are perceptions of people who wanted to be screened and now they are explaining to us why they wanted to participate in screening. Results must be presented as such. Examples of framing: "one of the most common reasons stated by respondents for participating in screening was....", "another important motivation for their participation was....", "the perception of the respondents who volunteered to be screened against those who don't was....", "underlying perceived benefits that compelled these adults to participate in screening included...."

Patients' knowledge of AF and stroke is an important factor that impacts how we interpret the results but I am not sure if it should be a theme of its own. It has nothing to do with the research question. It is perhaps best presented in other themes where relevant. Or at least provide it early in the results to provide the context for understanding the screen related results.

Table 1:

what is a deprivation score? What do the letters represent? Please clarify in footnote.

What was the participant's education level? This is important in putting the findings in context

	Male and female are sex not gender. Discussion The following sentence needs to be changed: “This may limit the relevance of our findings for understanding patient engagement in either opportunistic AF screening conducted as part of routine primary care, or potential future systematic AF screening programmes”. Your study is not supposed to have relevance for understanding patient engagement in opportunistic AF screening. It wasn’t designed for that and with the serious selection bias it is 100% unable to do that. Your study was supposed to explore reasons for participation in screening in those who have actually participated, so this sentence should very clearly and in a definitive way communicate this limitation. The justification provided for the selection bias in discussion is as follows: “Nevertheless, focusing on people who did engage has allowed us to understand the information needs and potential misconceptions of this population”. This justification is invalid because understanding the information needs and potential misconceptions of AF patients was not the objective of the study. While the sample size of this study is impressive for a qualitative study, the serious selection bias present warrants the need to interview more people from the general public to provide a balanced, accurate and representative report of AF patients’ perceptions Again, not sure what a deprivation score is. The first paragraph of discussion of findings discusses participants’ knowledge gaps and misconceptions about AF/Stroke. This was not the objective of your study and hence it is the least important finding of the study. I understand that in qualitative studies findings emerge that were not expected but they certainly do not deserve to make up 1/3rd of discussion of results. Was the potential barriers to screening explored at all? Theoretically what would have stopped these patients from wanting to participate? I am surprised barriers such as access and cost has not come up. Did the participants maybe knew about these contextual factors (for example they knew the screening if offered is going to be free for all in the UK). In discussion as well, you are pretending that your results are from a random sample of participants and then comparing it with
--	---

	findings of studies of the public support for screening: “The enthusiasm participants had for AF screening, and the weighting they attributed to the benefits versus the harms of screening, accord with other studies on patient experience of national screening programmes (43,44). There is widespread public support for screening even in survey scenarios in which there was no treatment for the screened condition (25,26), or if screening is not clinically recommended because participants are outside screening age thresholds”. Discussion must be around the reasons identified for participation in screening (the objective of your study) and whether they support/dismiss the reasons for screening participation reported by other studies. You cannot use these results to make any comments about public opinion of screening.
--	--

VERSION 1 – AUTHOR RESPONSE

Reviewer 1

No.	Comment	Response
1	Methods Data collection : Page 7 line 23-26: Please provide more details regarding the prompts used during interview. Only a general statement is provided. It would be appropriate to add the interview prompt schedule as an appendix, particularly as in the analysis section you refer to the themes being developed deductively from the interview schedule. It's therefore important we have more information about that schedule, to this understand the basis of the themes that are presented.	Thank you for this comment. We have added in the interview prompt guide as an appendix, and included a reference to this in the text “(see appendix)”
2	Analysis Page 7 Line 32: provide an explanation for the statement that ‘16 transcripts were selected for their relevance’. What do you mean by relevance please define? Page 7 Line 41: figure one. The figure does not transparently report at which point across the three interviews the final 16 were selected. Were the same number selected at each interview point? Or were there different numbers at different interview points? Arguably the patient experience is different at each interview point and this may have influenced their experience and the information they provided.	Thank you for these comments. We have expanded the text to specify what we meant by relevance, and to make clear the timepoints when interviews were selected. The reviewer’s comment here and below (comment 5) also suggests the description of our analysis was unclear and incorrectly infers we only analysed 6 interviews. In practice we analysed the whole dataset, using an initial 16 to establish the key themes which were explored through consensus meetings and subsequently developed through analysis of the whole dataset. To make this clear, we have amended the analysis text so it now reads: The initial analysis phase focused on 16 transcripts selected for their relevance to

		addressing the question and drawn from all three interview time-points (interview 1, 7; interview 3, 3; interview 3; 6) followed by analysis of the whole interview dataset. [Author 1] conducted coding, on paper and then supported by the software NVivo 12, with codes generated inductively from topics raised by participants and interview fieldnotes, and deductively from the interview schedule. We developed key themes through consensus meetings between [authors 1, 2, 7], and [author 1] then explored these within the wider dataset to establish the veracity of key themes and identify deviant cases, with the themes subsequently refined. The reviewer also makes an important point about the longitudinal interview design. We agree that the patient experience is likely to be different at each interview point. We provide further explanation of our approach in response to reviewer 2, comment 13.
3	Page 7 line 39: you refer to the themes being synthesised and aided in this process by reference to social science and health screen literature. Please provide references for the literature you used as the basis for managing the themes.	In our analysis we drew on a wide selection of literature. The key literature that helped our analysis is referenced in the discussion, and so rather than providing a very long list of references we have clarified in the text what the references addressed. This section now reads: “These themes were synthesised to understand shared views of screening participation, aided by reference to social science and health screening literature about participation in screening.”
4	Patient and public involvement Page 7 line 46: here there is a term ‘PPI representatives’. I can't see that defined previously if it hasn't been, please provide that here. The term may be particular to the UK	Thank you for alerting us that PPI is not a universal term. In the title to this section we defined what the acronym PPI is. To help non-UK readers, we have now also included a reference which defines the role of PPI representatives: “The SAFER programme is guided by four PPI representatives (29)”
5	Results Page 7 line 58: here the authors refer to completing 53 interviews. However previously in the analysis section you state that you completed 16 transcripts/interviews. This is very unclear. I note later in the analysis section you refer to the ‘rest of the data set’. If you mean that the data set was not	Thank you for this comment. The results we present are based on the whole dataset of 53 interviews: we initially analysed 16 interviews, and then explored this within the rest of the dataset. We have amended the text to

	the 16 transcripts but was the wider 53 interviews, then this needs to be much clearer in the analysis. Figure 1 should also reflect this process.	make this clear: please see comment 2 above for the new text. We note the reference to figure 1. The main purpose of this figure is to illustrate when interviews were conducted during the SAFER study process and to explain where interview participants were not interviewed or chose to withdraw. We are concerned that adding to the diagram information about the initial 16 analysed interviews would detract from the clarity of the diagram and may unhelpfully contribute to the inaccurate assumption that our analysis was based only on these 16 interviews and not the wider interview dataset. We also hope that amendments to the text have resolved this issue. To clarify the purpose of this figure, we have amended the title, so it now reads: [Figure 1: Flowchart of patient interviews in the screening process]
6	Page 7 line 58 and 59: please provide more demographic information Regarding the participants under a subheading labelled demographics . The inclusion of the results at this point is a little unclear, please provide another subheading to introduce the sentence / section beginning with 'engagement with AF screening'. The sentence from page 7 line 59 through to page 8 line 7 seems to be related to the following subheading on the importance of screening. It might be relevant to merge the two.	We agree providing demographic information on participants is helpful for the reader. As we are limited for space, this information is provided in Table 1. To make this clear to the reader, we have moved text references to Table 1 from the methods section to the results. Thank you for identifying that the results first paragraph is unclear. The purpose of this introductory paragraph is to introduce and summarise our results, which are then expanded in full in the following paragraphs. We have implemented the suggestion of a subheading titled 'summary'.
7	Discussion Page 11 line 58 to 60 and 12 line 3 to 4: this is an important discussion point and speaks to the relevance of this paper. The debate for AF screening is particularly strongly based around the need to screen and therefore engage hard to reach and vulnerable populations. These populations are not included in this study, It will be highly relevant to include some discussion regarding the learnings from this study as applied to the hard to reach / vulnerable populations. Can the factors that drive an already engaged (and relatively privileged group going by the demographics) inform engagement of hard-to-reach populations? I think the manuscript would be greatly enhanced by	Thank you. We agree it is important to explore the reasons why people do not engage or take part in AF screening, so much so that our next paper is on this topic. Given the importance of the issue, word count limits, and a desire not to address the issue in a tokenistic way, we have not sought to include this issue in the current paper.

	expanding on this point and truly contribute to the discussion on the broader issue of AF screening.	
2	Page 13 line 3: ‘...because clinical evidence on the benefit of managing screen detected AF is still uncertain’. This phrase speaks to the debate against screening in that screening can induce anxiety additional requirements for medical appointments and treatment and particularly amongst underserved populations this can be an ethical issue to consider. This phrase warrants a little more discussion.	Thank you. To clarify, when we refer to the uncertainty of managing screen detected AF, we refer to the broad question of whether screen-detected AF has the same clinical consequences as AF detected through present routes (e.g. following GP appointment for AF-related symptoms), and therefore whether screen-detected AF patients would benefit from the same treatment as other AF patients. We agree with the reviewer that the harms of screening are a key consideration, and we intend to explore this point in future papers. In this paper, we have updated the text to make this explicit. The text now reads: “... and because clinical evidence on the benefit of managing screen-detected AF is still uncertain (18). As with all screening, benefits must be weighed against harms (64).

Reviewer 2

No.	Comment	Response
1	The study addresses an important research question and uses an appropriate methodology to do so. Currently, however, the paper is written as if results are perceptions of the general public, they are not. They are perceptions of people who wanted to be screened and now they are explaining to us why they wanted to participate in screening. Most of my suggestions in the attached document are about correctly framing the interesting findings you have so they are not misleading to the reader. I am looking forward to reading the revised version of this manuscript.	Thank you. To clarify, the paper is about the perceptions of people who were randomly invited to take part in a study about screening via their GP practice, and who chose to take part in the offer of screening. We hope that from the changes we have made in response to the reviewer’s related comments below we have clarified this.
2	Abstract Under design n is stated as 53 but under results n is reported to be 23. In the manuscript it becomes clear that 53 is the number of interviews and 23 is the number of participants (multiple	Thank you for identifying this confusion. We have clarified this by omitting the number of interviews in the design section, and moving it to the participants section, which now reads: “Participants: 23 people (n=53 interviews) taking part [...]”

	interviews per patient) but this is not clear in the abstract and is confusing.	
3	Objective is misleading: “We aimed to explore why people take part in AF screening”. You aimed to explore why people who took part in AF screening did so.	We have amended the text as follows: “why AF screening participants took part in the screening”
4	Under design, “Interviews explored participants’ experience of and attitudes towards AF screening.” Is reiteration of the objective. Suggest using this space to provide more detail on the methods of the paper.	Thank you for identifying this repetition. We have omitted this sentence and expanded the analysis sentence so it now reads: We thematically analysed and synthesised this data to understand shared views of screening participation.
5	Results is misleading: “participants were supportive of screening for AF, describing it as a “good thing to do”. In those who took part in screening, one of the reasons stated was that it is a good thing to do.	We have amended this sentence as follows: Participants were supportive of screening for AF, explaining their participation in screening as a ‘good thing to do’.
6	Introduction Reads well and is appropriately cited.	Thank you for this comment.
7	Methods Please make sure all abbreviations are explained the first time they appear in the manuscript even the commonly used ones such as ECG and GP.	Thank you for highlighting this. In the text we have explained what the terms GP and ECG mean.
8	Please provide more information on the steps involved in thematic analysis. If word limit is the issue, such details can be provided as appendix.	Thank you for this. In the analysis section we cover who did the thematic analysis and the materials used, the steps we undertook to break the data down into codes, and how we synthesised it back together into themes to address the research question. We are unclear what further information the reviewer would wish to see included. We would be happy to provide further text on specific aspects, if requested.
9	Please attach your coding tree or code book as appendix.	Our thematic analysis, whilst involving coding, did not result in a coding tree. We note that whilst a coding tree is used in some forms of thematic analysis, it is not a requirement of the overall approach. In this context, providing a tree/code book would give an inaccurately neat portrayal of our analytic coding process, including overstating the significance of coding to thematic synthesis in our approach. As described in the text, coding was the first phase in our analytic process, followed by multiple stages of theme development.

10	You cannot use the word patient. Participants in your study and all those who will ever go for screening are not patients because they have not yet been diagnosed. This happens often in the manuscript e.g: “Patients can be expected to be unfamiliar with AF”, “eligible patients were aged 65 years and over, not taking anticoagulant medication, and were neither on a palliative care register nor living in a residential (care) home”. Participant, the public, and adults, are good replacements. Please correct throughout the manuscript.	Thank you for highlighting this. We have changed references to ‘patients’ to ‘the public’ throughout. We have retained a few references to ‘patients’ where it refers to a concept (e.g. ‘good patient’), a clinical encounter (e.g. “clinicians are recommended to assess patients’ pulse rhythm as part of NHS health checks’) or when referring to reference text that addresses patient experience.
11	Please provide a sample of the interview guide in the appendix	Thank you for this comment. We have added in the interview prompt guide as an appendix, and included a reference to this in the text “(see appendix)”
12	More information is required on the methods:  • when did interviews stop? Was the sample size prespecified or did the interviews continue until the point of saturation? How was saturation defined? • How was the interview guide designed? Based on author’s clinical experience? Review of the literature? Adaptation of an interview guide from another study? • Was there a pilot phase to improve the interview guide? And the interview process? • Was the interview guide revised iteratively as more interviews were conducted? • Was the analysis done iteratively or at the end? And why. 	Thank you for these suggestions: we address each of these questions in our response below. Sometimes we have not suggested changes to the text and provide information only in our response, and sometimes our changes to the text are brief. We recognise that methods sections can never provide a total account of the research conducted, in part due to the limitations of word count. We advocate for methods sections to contain sufficient material for readers to have a clear understanding of how data were collected and analysed to enable them to make a judgement on the robustness of the analysis and the veracity of the results presented. We also consider that including superfluous information, often simply to meet reporting guidance, may inhibit reader understanding, thus detracting from the purpose of the section. Stopping interviews. Whilst we contest the utility of the term saturation in this context (see comment 18) we agree it can be helpful for readers to understand how interviews were concluded. We have therefore included the text: Interviews were conducted in 2019, and continued until we had reached sufficient information power (29) to have a meaningful dataset. Interview guide (design and iteration). Our initial topic guide was developed by the research team and drew on our experience from previous interview studies, prior awareness and

		reading about the topic and adjacent ones (e.g. screening generally) to inform the questions we asked. The questions were refined as interviews progressed to reflect interviewees' responses (e.g. if topics felt 'awkward' to ask or respond to, we reframed how they were approached) and iteratively adapted to reflect areas of interest raised by participants. We have included the sentence: The topic guide was designed to reflect our research question and was refined and iteratively adapted as interviews progressed to accommodate areas of interest raised by participants. Piloting. We did not pilot the topic guide, although as described above it was reviewed and revised by the research time throughout the interview process. We did not officially pilot the interview process, although Author 1 did conduct an informal interview with a person known to the research team who had AF as part of the preparation for the interviews. As these are things we did not do, we have not amended the text in relation to these points. Analysis start. Formal analysis started following the completion of interviews for practical reasons (the interview schedule was intense to fit interviews in with the trial programme, and therefore there was not time to concurrently analyse). However, we wrote comprehensive fieldnotes following each interview in which we reflected on the interview topics, so in practice our analytic process started following the first interview. In response to the reviewer's helpful identification of our omission about fieldnotes (see 18, below) we have amended the analysis text to make explicit we did so.
13	What was the reason for interviewing participants 3 times. You have not assessed changes in perception/understanding/willingness of participants throughout time, and have treated the interviews as if they are each independent of one another, so why the longitudinal design?	Thank you for this comment. In this study, we were not seeking to understand how any one individual's views on screening participation changed over time but we were interested in exploring the perceptions of participants at different stages of the screening process. We therefore conducted interviews at three time points. Significantly for this study, participants helpfully commented at all three interview points on why they had chosen to take part, as evidenced in the quotations. To make this more explicit, we have clarified the interview codes that follow the quotations (please see response to comment 15).
14	Results Actual mean interview duration must be reported.	Thank you. We have now amended this text so it reads: Interviews lasted on average 30 minutes each (range 10-90 minutes)

15	It is not clear what the codes in the brackets represent. Assuming they refer to the participants ID, I am not sure if having them would help the reader in any way to better contextualize or understand the quote. Basic information such as (Male, 65) may be more helpful.	Thank you for making us aware that the codes we use are not clear. We have provided a clarifying sentence in the introduction to the results section: "Quotations in the text are followed by the participant's interview ID number (1-53), practice code (A-E) and interview phase (1-3)" The reviewer also asks about the utility of these codes. We report the participants' ID and a practice ID code primarily to facilitate trust with the reader about the robustness of our results. We hope that ID codes help the reader to understand, for example, where a participant's account is used multiple times and to see the spread of viewpoints between practices. We chose not to report participants' age and sex in the codes as our sample was too small for these to be useful analytic tools.
16	Quotations are usually italicized. I refer to the editors on if they should be in this paper. It is not clear which of the titles represent the themes and which are the subthemes (if there is any). Numbering the themes usually helps with this. Some of the titles are in red, I am not sure why. Are these the main themes? It is very likely that they will all show up as black in the final version so this is not the best way to distinguish themes from subthemes.	We have amended the formatting of the text as suggested, with text in black and headers numbered/lettered to indicate heading hierarchy.
17	Theme names should be more descriptive and meaningless even without the text. Titles such as "low risk", "screening", "AF" are not helpful.	Thank you for suggesting this, although we suspect you mean meaningful rather than meaningless. We have reviewed all our results section subheadings and have amended the titles as follows:  Clinically recommended screening Reassurance of screening Views about AF Views about AF screening Low-risk screening
18	The point of reporting guidelines such as COREQ is to ensure standardization of reporting and to ensure that all the important information about a study has been reported. You cannot have NA in your checklist, all information in COREQ must be reported in your manuscript. I am unable to find information on field notes, information on how many data coders	Thank you for this comment. In line with the reviewer's suggestion to ensure that the information we report is easy to find we have updated the COREQ form to include title/subtitles. (These supplement the page numbers to meet the form requirements). The reviewer notes that we have reported 'N/A' on some answers. We agree that standardised reporting is helpful. However, reporting guidelines reflect interpretations of qualitative methods which are not universally shared, and

	coded the data and information on interviewer's relationship with the participant, even though the authors indicate having reported them in the COREQ checklist. Page numbers change throughout the publication process, instead of page numbers it might be helpful to provide the title/subtitle under which the info appears. For example page 7 is supposed to have information on whether the guide was pilot tested but this information is not there	therefore it may be appropriate to report 'not applicable'. (We also note that the COREQ guidance permits the use of 'n/a' in its guidance). For example, item no. 23 'data saturation' is derived from Grounded Theory and is irrelevant to qualitative analysis that does not use either Grounded Theory or a grounded theory inspired approach. The reviewer raises several specific concerns, and we respond to each as follows: We acknowledge that references to field notes were inadvertently missing. To address this, we have made explicit reference to fieldnotes in the data collection and analysis section. The text now reads: Data collection: Fieldnotes were completed following each interview. Analysis: [Author 1] conducted coding, on paper and then supported by the software NVivo 12, with codes generated inductively from topics raised by participants and interview fieldnotes, and deductively from the interview schedule. We note that information on coders is provided in the analysis section sentence "[Author 1] conducted coding, [...]" In the paper we did not provide direct reference to our relationship with the participant. However, we do state how participants were selected (i.e. purposively to meet sampling criteria, via the trial) from which we hope readers will anticipate that our relationship with the participant was as an interviewer for the trial. COREQ item 17 asks "Were questions, prompts, guides provided by the authors? Was it pilot tested?". We provide information on the interview topic guide in the 'data collection' section. We did not formally pilot the guide so do not report this. (for further response on the piloting of the interview guide please see also comment 12).
19	I am not sure if dividing the results on screening in general and AF screening specifically is a good idea. It is confusing, at times feel repetitive.	We think it is important to be explicit about participants' general views on screening and those they held about AF screening. This is particularly important because this is a new screening programme. The division allows us to demonstrate, for example, that participants drew on their pre-existing views of screening to assess their participation in AF screening.
20	The fact that the people who were interviewed are those who volunteered to join a screening	Thank you for raising this comment. We agree that it is important to acknowledge that the screening offered to participants was offered as part of the research study they were participating in. This

	program, needs to be emphasized throughout the paper and more clearly in discussion. It is not so much an issue that they were part of a prior research study but that they were part of a screening research study. Results is written as if these are perceptions of general public, they are not. They are perceptions of people who wanted to be screened and now they are explaining to us why they wanted to participate in screening. Results must be presented as such. Examples of framing: “one of the most common reasons stated by respondents for participating in screening was....”, “another important motivation for their participation was....”, “the perception of the respondents who volunteered to be screened against those who don’t was....”, “underlying perceived benefits that compelled these adults to participate in screening included....”	was something we sought to be attentive to during data collection, analysis and writing. We are explicit about this throughout the paper, making comment about it including in the abstract¹, strengths and limitations opening section², introduction³, methods⁴, and opening both the results⁵ and discussion⁶ section with reference to it. We agree that the reviewer’s suggested changes to the results would further reinforce this message. However, the changes proposed also seem likely to interrupt the presentation of our discussion, hindering the clarity and thus the reader’s assessment of our findings, and significantly, due to the limits of the word count, would reduce what we could report. We have therefore not made changes to the text. ¹ “Design. Semi-structured longitudinal interview study of participant engagement in the SAFER study (Screening for Atrial Fibrillation with ECG to Reduce stroke).” ² “We report the views of people taking part in AF screening as part of a research study, limiting the relevance of our findings for understanding public engagement in either opportunistic AF screening conducted as part of routine primary care or potential future systematic AF screening programmes” ³ “We aimed to explore why participants in SAFER, an AF screening study, opted to take part.” ⁴ Please see ‘design and participants’ section ⁵ “Engagement with AF screening within the SAFER study was driven by several inter-connected considerations [...]” ⁶ Summary. “We explored the reasons why people took part in AF screening through interviews with SAFER study participants.” Strengths and limitations. “It is likely that our results reflect the delivery of SAFER as a research study [...]”
21	Patients’ knowledge of AF and stroke is an important factor that impacts how we interpret the results but I am not sure if it should be a theme of its own. It has nothing to do with the research question. It is perhaps best presented in other themes where relevant. Or at least provide it early in the results to provide the context for understanding the screen related results.	We respectfully disagree that patients’ knowledge of AF and stroke is not related to our research question ‘why people chose to take part in AF screening’. Participants’ enthusiasm for participating in screening despite their limited understanding of the screened condition is a key finding and helps to demonstrate how screening motivation may be related more to the perceived importance of engaging with screening than to the specific screened condition – a point we elaborate in the discussion. We have therefore not made any changes to the text.

22	Table 1 what is a deprivation score? What do the letters represent? Please clarify in footnote. What was the participant's education level? This is important in putting the findings in context Male and female are sex not gender.	Thank you for identifying these omissions and the mistake: we have updated the label for sex and have provided clarifying footnotes about the deprivation score and practice letters. These read: The deprivation score is taken from National General Practice Profiles (71), using the English Indices of Deprivation to calculate the Index of Multiple Deprivation, which provides “an overall measure of deprivation experienced by people living in an area” (71,72) ² Practice name pseudonym We agree that education level could be a useful criterion for understanding why people chose to take part in screening. However, this data was not collected as part of the trial, and therefore we cannot report it.
23	Discussion The following sentence needs to be changed: “This may limit the relevance of our findings for understanding patient engagement in either opportunistic AF screening conducted as part of routine primary care, or potential future systematic AF screening programmes”. Your study is not supposed to have relevance for understanding patient engagement in opportunistic AF screening. It wasn't designed for that and with the serious selection bias it is 100% unable to do that. Your study was supposed to explore reasons for participation in screening in those who have actually participated, so this sentence should very clearly and in a definitive way communicate this limitation.	We agree that the main aim of our study was to explore reasons why SAFER participants chose to take part in AF screening. We respectfully disagree that this means that we should not report the relevance of our findings to understanding patient engagement in opportunistic AF screening, providing the scope of our findings is clearly established (i.e., that they are derived from a systematic screening trial). For example, both opportunistic and systematic screening programmes are seeking to screen participants for AF and given that both use a similar population (the public) we can expect that participants in either approach may also have limited understanding of AF as we found with our study population. The need to establish the relevance of our findings with opportunistic AF is particularly acute given the dearth of studies that report participant views of any type of AF screening and the prevalence of opportunistic screening. We have not amended the text.
24	The justification provided for the selection bias in discussion is as follows: “Nevertheless, focusing on people who did engage has allowed us to understand the information needs and potential misconceptions of this population”. This justification is invalid because understanding the information needs and potential misconceptions of AF patients was not the objective of the study. While the sample size of this study is impressive for a qualitative study, the serious selection bias present warrants the need to interview more people from	Thank you for this comment, and the opportunity to explain our perspective. Our study purpose was to understand why people take part in AF screening. As part of that research, we found that an awareness of AF was not a key factor for people taking part (i.e., participants took part despite knowing little about AF). This research is of significance to those delivering screening programmes, particularly in considering how best to ensure that participants are appropriately informed about the screening and able to give informed consent. More broadly, we suggest that the concept of bias, with its quantitative heritage, can itself be contentious in qualitative research. Further, a study

	the general public to provide a balanced, accurate and representative report of AF patients' perceptions	of the general population would not prevent similar 'biases' from entering the study. For example, if study participants were asked to share their views on screening without being offered it, we would have to acknowledge that their views were of a hypothetical situation, and might not therefore reflect their decision making when presented with a 'real-life' screening opportunity. We have not made changes to the text.
25	Again, not sure what a deprivation score is.	Thank you. Please see our response to comment 22 where we clarify this.
26	The first paragraph of discussion of findings discusses participants' knowledge gaps and misconceptions about AF/Stroke. This was not the objective of your study and hence it is the least important finding of the study. I understand that in qualitative studies findings emerge that were not expected but they certainly do not deserve to make up 1/3rd of discussion of results.	This comment has similarity with the reviewer's concern about our focus on this topic in the results section and our justification for not amending the text matches our response there. Please see comment 21.
27	Was the potential barriers to screening explored at all? Theoretically what would have stopped these patients from wanting to participate? I am surprised barriers such as access and cost has not come up. Did the participants maybe knew about these contextual factors (for example they knew the screening if offered is going to be free for all in the UK)	Thank you for this comment. We agree that it is important to explore why people may not take part in screening, or what may inhibit them from taking part, and we plan on addressing this in a subsequent paper. Due to the remit of the current paper, we have chosen not to refer to this point in this paper where by necessity it would only be a nominal reference to a significant topic. Please also see our response to reviewer 1 (comment 7).
28	In discussion as well, you are pretending that your results are from a random sample of participants and then comparing it with findings of studies of the public support for screening: "The enthusiasm participants had for AF screening, and the weighting they attributed to the benefits versus the harms of screening, accord with other studies on patient experience of national screening programmes (43,44). There is widespread public support for screening even in survey scenarios in which there was no treatment for the screened condition (25,26), or if screening is not clinically recommended because participants are outside screening age thresholds".	Thank you for this comment. As you note, we are explicit in our reference to these studies that they are about the public - we have not sought to hide this distinction. We would respectfully disagree that our findings do not have relevance to published studies about the enthusiasm for screening: we consider it useful to demonstrate to readers how our findings parallel these public screening studies and therefore to contribute to a wider discussion about how the public recognise and weigh harms and benefits in screening. More broadly, we consider these topics within the scope of a discussion section and contend that these points do contribute to our understanding of why people take part in AF screening (please see also previous comments 20, 23). We have not made changes to the text.

Discussion must be around the reasons identified for participation in screening (the objective of your study) and whether they support/dismiss the reasons for screening participation reported by other studies. You cannot use these results to make any comments about public opinion of screening.	
---	--

VERSION 2 – REVIEW

REVIEWER	Gwynn, Josephine University of Sydney - Mallett Street Campus, charles perkins centre
REVIEW RETURNED	11-Nov-2021

GENERAL COMMENTS	There are still serious confusions about participant inclusions, numbers, and analysis. Clarification of these issues in the manuscript (this was a bit clearer in the response to reviewers but needs to translate to the paper) will then enable results/discussion to be read more confidently. Appear well written but need to be confident of the data. ABSTRACT Participant number statement in Abstract still not clear. Remove the n=53 statement as not helpful here. In abstract we just need to know who participated in this study. Would Atrial Fibrillation be a key word? ANALYSIS Confusion still reigns here. I think this is in part because you are not contextualising the analysis process clearly. Your first statement in the analysis section is the mention of 16 transcripts selected for their relevance which implies 16 participants. you then proceed to talk about three interview time points, and then an analysis of the whole interview data set by which stage the reader is totally confused! It appears that the first sentence of the results section provides the context that is needed before you start talking about the analysis process in the analysis section. So I suggest either moving all of the information about the numbers of transcripts and analysis process to the results to follow the statement about the 53 interviews OR move the first sentence of results into the analysis section as the first sentence there so that you can contextualise the analysis process. The analysis process itself is not clear. When I looked at the interview guide, no question is asked about AF screening in Interview 1. Did the 7 people included in this study from Interview 1 mention AF screening unprompted? no SAFER screening occurred before Interview 1. Clarify. Authors state an initial analysis phase of 16 transcripts. Were these from 16 participants? Or fewer if a number of transcripts exist per participant? The latter not discussed, but could be assumed as presumably the same participants were eligible at each interview point. Note 23 participants (is this 23 transcripts?) were stated in the abstract and again in results. Then the authors state that they followed up with analysis of whole interview dataset
--

	(n= 53 interviews). Why were all analysed if the initial interviews were selected for relevance (implying the remainder were not relevant). Quite confusing. In your response to reviewers table you do address this a little clearer than in the paper and suggest you use some of those explanations in the paper. I presume each transcript was a new participant. That you didn't include several interviews from the one person? If you did then this need explaining Edit note:... Interview 2, 3: Interview 3, 6. Where is Figure 1? If you used NVIVO (as indicated in author response doc) then you must indicate this in the analysis section. Reference to literature here requires at least a couple of key articles - i note your response to reviewers but that is insufficient in an academic piece of work. RESULTS Now I realise another point of confusion - the section B on Screening refers to any screening not AF screening. This is not clear, and has only become apparent to me from reading the interview guide now included. For an article with the title AF screening the natural assumption is that this is the topic of the manuscript. So relabel the section on 'importance of screening' to clearly state what is included ie screening in general! AF screening is in Section C. The manuscript title may need a rethink as well. Had participants whose information informed the General Screening section (which was gathered during Interview 1 according to guide) experienced any screening... or was this factor not considered? If the dataset includes the responses from those prior to screening and then after screening, then you have 2 datasets surely? Views may change between before and after. This all needs more explanation. Did all participate in AF screening? You only conducted 20 interviews at Interview 3 (or 6 according to the Analysis section)... so even if all the transcripts from these people are included you are still missing 3 participants (I presume all 23 contributed to all the results provided as you have not indicated otherwise). The veracity of the results can be more confidently established when the above is clarified. DISCUSSION The above impacts on the Discussion. This section seems well written however this can only be assessed once the outstanding queries about the data management and inclusions have been addressed.
--	---

REVIEWER	Salmasi, Shahrzad Collaboration for Outcomes Research and Evaluation (CORE), Faculty of Pharmaceutical Sciences, Faculty of Pharmaceutical Sciences, The University of British Columbia
-----------------	---

REVIEW RETURNED	03-Nov-2021
-------------

GENERAL COMMENTS	no further comments
---------------------

VERSION 2 – AUTHOR RESPONSE

Reviewer 1

No.	Comment	Response
1	There are still serious confusions about participant inclusions, numbers, and analysis. Clarification of these issues in the manuscript (this was a bit clearer in the response to reviewers but needs to translate to the paper) will then enable results/discussion to be read more confidently. Appear well written but need to be confident of the data.	Thank you for reviewing the manuscript again. We note your concerns and hope the changes we have made, outlined below, in response to your comments address these.
2	Participant number statement in Abstract still not clear. Remove the n=53 statement as not helpful here. In abstract we just need to know who participated in this study.	Thank you for identifying this. As suggested, we have omitted this text. It now reads: “Participants: 23 people taking part in the SAFER study first feasibility phase.”
3	Would Atrial Fibrillation be a key word?	Thank you for this suggestion: we have swapped ‘cardiovascular disease’ for ‘atrial fibrillation’.
4	ANALYSIS Confusion still reigns here. I think this is in part because you are not contextualising the analysis process clearly. Your first statement in the analysis section is the mention of 16 transcripts selected for their relevance which implies 16 participants. you then proceed to talk about three interview time points, and then an analysis of the whole interview data set by which stage the reader is totally confused!. It appears that the first sentence of the results section provides the context that is needed before you start talking about the analysis process in the analysis section. So I suggest either moving all of the information about the numbers of transcripts and analysis process to the results to follow the statement about the 53 interviews OR move the first sentence of results into the analysis section as the first sentence there so that you can contextualise the analysis process.	Thank you for identifying this confusion and the suggested solution. In line with this, and so that readers have the information on interviews in advance of the analysis section, we have moved the following sentences from the results section to the data collection section where the interviews are first discussed: “We completed 53 interviews with 23 participants (interview 1 n=23, interview 2 n=10, interview 3 n=20). Table 1 lists participants’ characteristics.”

5	The analysis process itself is not clear. When I looked at the interview guide, no question is asked about AF screening in Interview 1. Did the 7 people included in this study from Interview 1 mention AF screening unprompted? no SAFER screening occurred before Interview 1. Clarify. Authors state an initial analysis phase of 16 transcripts. Were these from 16 participants? Or fewer if a number of transcripts exist per participant? The latter not discussed, but could be assumed as presumably the same participants were eligible at each interview point. Note 23 participants (is this 23 transcripts?) were stated in the abstract and again in results. Then the authors state that they followed up with analysis of whole interview dataset (n= 53 interviews). Why were all analysed if the initial interviews were selected for relevance (implying the remainder were not relevant). Quite confusing. In your response to reviewers table you do address this a little clearer than in the paper and suggest you use some of those explanations in the paper. I presume each transcript was a new participant. That you didn't include several interviews from the one person? If you did then this need explaining Edit note:..... Interview 2, 3: Interview 3, 6.	Thank you for these comments. We have responded to each below, including text modifications to make our analysis approach clearer. Interview 1 The interviews were semi-structured, and participants did, unprompted, refer to the AF screening in their first interview. More broadly, across all three interviews participants expressed ideas about AF screening that were relevant to understanding their views about participating. To make this clear, we have amended the data collection section, so it now reads: “Interviews were semi-structured and used a flexible topic guide exploring experiences of, and attitudes towards, screening in general and AF screening in particular (see supplemental material). These topics were explored by participants throughout all three interviews.” Initial analysis phase/16 transcripts/23 participants 23 participants were interviewed up to three times and in total we completed 53 interviews (not all interviewees took part in all interviews, and we state this in the data collection section). We collated the 53 interviews into a single dataset, and from these chose 16 interviews, drawn from across the three interview timepoints, to conduct our initial analysis with. The 16 interviews included some from the same participant where this was useful. To help clarify this, we have amended the text, and it now reads: Interviews were collated, and the initial analysis phase focused on 16 transcripts selected for their relevance to addressing the question; these were drawn from all three interview time-points (interview 1, 7 transcripts; interview 2, 3 transcripts; interview 3, 6 transcripts; with some participants represented more than

		once). This initial analysis was followed by analysis of the whole interview dataset (n=53). Analysing the whole dataset All the interviews we collected were useful and helped us to understand why participants took part in AF screening. We conducted the initial analysis with a manageable subset of the whole dataset to gain detailed insight into why participants engaged in the screening. As we state in the analysis section of the manuscript, we then expanded our analysis to the whole dataset to 'establish the veracity of key themes and identify deviant cases, [and subsequently refine these themes].'
6	Where is Figure 1?	We are sorry that you could not review this figure. It was included in the submission pdf, after the clean copy version of the manuscript.
7	If you used NVIVO (as indicated in author response doc) then you must indicate this in the analysis section.	We did use NVivo, and reference to our use of the software is stated in the analysis section, in the sentence: "SH conducted coding, on paper and then supported by the software NVivo 12, with codes generated inductively from topics raised by participants and interview fieldnotes, and deductively from the interview schedule."
8	Reference to literature here requires at least a couple of key articles - i note your response to reviewers but that is insufficient in an academic piece of work.	Thank you. We have now included the key citations in the text as follows: "These themes were synthesised to understand shared views of screening participation, aided by reference to social science and health screening literature about participation in screening [30–38] ."
9	RESULTS Now I realise another point of confusion - the section B on Screening refers to any screening not AF screening. This is not clear, and has only become apparent to me from reading the interview guide now included. For an article with the title AF screening the natural assumption is that this is the topic of the manuscript. So relabel the section on	Thank you for these comments: we address each of the topics raised below. Section B: Title This section addresses participants' general views of screening, which they drew on to explore the new concept of atrial fibrillation screening. Thank you for identifying that

	'importance of screening' to clearly state what is included ie screening in general! AF screening is in Section C. The manuscript title may need a rethink as well. Had participants whose information informed the General Screening section (which was gathered during Interview 1 according to guide) experienced any screening... or was this factor not considered? If the dataset includes the responses from those prior to screening and then after screening, then you have 2 datasets surely? Views may change between before and after. This all needs more explanation. Did all participate in AF screening? You only conducted 20 interviews at Interview 3 (or 6 according to the Analysis section)... so even if all the transcripts from these people are included you are still missing 3 participants (I presume all 23 contributed to all the results provided as you have not indicated otherwise). The veracity of the results can be more confidently established when the above is clarified.	this section title was not clear. We have now re-labelled this section "The importance of screening in general". Section B: experience of screening Participants did have experience of screening. We state this at the start of section B: 'Participants were familiar with national screening programmes, often referring to their own or a relative's prior experience of taking part in screening.' Interviews before and after screening All interview participants took part in the screening. We spoke to interviewees after they had respectively: 1) received information about the study, and 2) received information about the screening and 3) taken part in screening (we state this is in the data collection section). Whilst it would have been interesting to explore how views on screening participation changed over time, this is not what our analysis looked at. Instead, we wanted to know more holistically why participants had decided to take part in screening. Participants explored aspects of this at each of the interview points and we therefore drew on the whole dataset to address the research question. Participating in interviews All interview participants took part in the screening, and 20 of these took part in interview 3. Three of the 23 participants did not take part in the third interview: one had withdrawn from the interviews after the first interview, one declined the interview invitation, and one did not reply to the invitation. Figure 1 provides a diagram of the interview flow (we appreciate that you did not have opportunity to review this) and we state in the data collection section that not all participants took part in all interviews.
10	DISCUSSION The above impacts on the Discussion. This section seems well written however this can only be assessed once the outstanding queries about the	Thank you for this comment. We believe that the aforementioned revisions remain consistent with the text in the discussion which we have not amended.

	data management and inclusions have been addressed.	
--	---	--

Reviewer 2:

No further comments.